# The Pilot Study of Immunogenicity and Adverse Events of a COVID-19 Vaccine Regimen: Priming with Inactivated Whole SARS-CoV-2 Vaccine (CoronaVac) and Boosting with the Adenoviral Vector (ChAdOx1 nCoV-19) Vaccine

**DOI:** 10.3390/vaccines10040536

**Published:** 2022-03-30

**Authors:** Surakameth Mahasirimongkol, Athiwat Khunphon, Oraya Kwangsukstid, Sompong Sapsutthipas, Mingkwan Wichaidit, Archawin Rojanawiwat, Nuanjun Wichuckchinda, Wiroj Puangtubtim, Warangluk Pimpapai, Sakulrat Soonthorncharttrawat, Asawin Wanitchang, Anan Jongkaewwattana, Kanjana Srisutthisamphan, Daraka Phainupong, Naphatcha Thawong, Pundharika Piboonsiri, Waritta Sawaengdee, Thitiporn Somsaard, Kanokphon Ritthitham, Supaporn Chumpol, Nadthanan Pinyosukhee, Rattanawadee Wichajarn, Panadda Dhepakson, Sopon Iamsirithaworn, Supaporn Phumiamorn

**Affiliations:** 1Medical Life Sciences Institute, Department of Medical Sciences, Ministry of Public Health, Nonthaburi 11000, Thailand; athiwat.k@dmsc.mail.go.th (A.K.); nuanjun.w@dmsc.mail.go.th (N.W.); wiroj.p@dmsc.mail.go.th (W.P.); warangluk.p@dmsc.mail.go.th (W.P.); sakulrat.p@dmsc.mail.go.th (S.S.); penpitcha.t@dmsc.mail.go.th (N.T.); pundharika.p@dmsc.mail.go.th (P.P.); waritta.s@dmsc.mail.go.th (W.S.); nadthanan.p@dmsc.mail.go.th (N.P.); rattanawadee.w@dmsc.mail.go.th (R.W.); panadda.d@dmsc.mail.go.th (P.D.); 2Institute of Dermatology, Department of Medical Services, Ministry of Public Health, Bangkok 10400, Thailand; oraya@inderm.go.th (O.K.); mingkwanwichaidit@inderm.go.th (M.W.); daraka@inderm.go.th (D.P.); 3Institute of Biological Products, Department of Medical Sciences, Ministry of Public Health, Nonthaburi 11000, Thailand; sompong.s@dmsc.mail.go.th (S.S.); thitiporn.s@dmsc.mail.go.th (T.S.); kanokphon.r@dmsc.mail.go.th (K.R.); supaporn.c@dmsc.mail.go.th (S.C.); supaporn.p@dmsc.mail.go.th (S.P.); 4National Institute of Health, Department of Medical Sciences, Ministry of Public Health, Nonthaburi 11000, Thailand; archawin.r@dmsc.mail.go.th; 5Virology and Cell Technology Research Team, National Center for Genetic Engineering and Biotechnology (BIOTEC), National Science and Technology Development Agency (NSTDA), Pathum Thani 12120, Thailand; asawin.wan@biotec.or.th (A.W.); anan.jon@biotec.or.th (A.J.); 6Epidemiology Division, Department of Disease Control, Ministry of Public Health, Nonthaburi 11000, Thailand; kanjana.sri@biotec.or.th (K.S.); iamsiri@yahoo.com (S.I.)

**Keywords:** homologous, heterologous, COVID-19 vaccine, heterologous prime-boost, immunogenicity, inactivated, adenoviral vector

## Abstract

In response to the SARS-CoV-2 Delta variant, which partially escaped the vaccine-induced immunity provided by two doses of vaccination with CoronaVac (Sinovac), the National Vaccine Committee recommended the heterologous CoronaVac-ChAdOx1 (Oxford–AstraZeneca), a prime–boost vaccine regimen. This pilot study aimed to describe the immunogenicity and adverse events of the heterologous CoronaVac-ChAdOx1 regimen, in comparison with homologous CoronaVac, and homologous ChAdOx1. Between May and August 2021, we recruited a total of 354 participants from four vaccination groups: the CoronaVac-ChAdOx1 vaccinee (*n* = 155), the homologous CoronaVac vaccinee (*n* = 32), the homologous ChAdOx1 vaccinee (*n* = 47), and control group of COVID-19 patients (*n* = 120). Immunogenicity was evaluated by measuring the level of IgG antibodies against the receptor-binding domain (anti-SRBD) of the SARS-CoV-2 spike protein S1 subunit and the level of neutralizing antibodies (NAbs) against variants of concern (VOCs) using the plaque reduction neutralization test (PRNT) and pseudovirus neutralization test (pVNT). The safety profile was recorded by interviewing at the 1-month visit after vaccination. The anti-SRBD level after the second booster dose of the CoronaVac-ChAdOx1 group at 2 weeks was higher than 4 weeks. At 4 weeks after the second booster dose, the anti-SRBD level in the CoronaVac-ChAdOx1 group was significantly higher than either homologous CoronaVac, the homologous ChAdOx1 group, and Control group (*p* < 0.001). In the CoronaVac-ChAdOx1 group, the PRNT_50_ level against the wild-type (434.5 BAU/mL) was the highest; followed by Alpha variant (80.4), Delta variant (67.4), and Beta variant (19.8). The PVNT_50_ level was also found to be at its highest against the wild-type (432.1); followed by Delta variants (178.3), Alpha variants (163.9), and Beta variant (42.2), respectively. The AEs in the CoronaVac-ChAdOx1 group were well tolerated and generally unremarkable. The CoronaVac-ChAdOx1 heterologous regimen induced higher immunogenicity and a tolerable safety profile. In a situation when only CoronaVac-ChAdOx1 vaccines are available, they should be considered for use in responding to the Delta variant.

## 1. Introduction

In early 2021, the COVID-19 vaccines were unavailable and inaccessible to low- and middle-income countries. Only two COVID-19 vaccines were available in the first half of 2021 in Thailand, inactivated whole-virus vaccine CoronaVac [1] (Sinovac Biotech, Beijing, China) and viral vector vaccines ChAdOx1 nCoV-19 vaccine (AstraZeneca, Oxford, United Kingdom) [2]. A heterologous prime–boost vaccination regimen induces better immunogenicity for various vaccines [3] and is regularly practiced in routine vaccination programs, such as the mixing of various influenza vaccines in the influenza vaccination program.

The SARS-CoV-2 Delta variant began its outbreak in early 2021 in India, where the Delta variant unfolded into a devastating health crisis with more than 400,000 reported death cases. However, an estimated number of deaths might be underreported since the Delta variant has high transmissibility and worse severity due to vaccine escape capability [4]. Public Health England, the US Center for Disease Control, and the World Health Organization recognized the Delta variant as a variant of concern [5]. The WHO published a global alert to announce that the Delta variant was becoming the globally dominant SARS-CoV-2 variant in mid-2021 and suggested accelerating vaccination coverage in response to the Delta variant [6].

The Delta variant was first detected in Thailand in May 2021 and became the dominant variant in the first week of August 2021 [7]. The Ministry of Public Health of Thailand launched a COVID-19 vaccination campaign using heterologous prime-boosted CoronaVac and ChAdOx1 nCoV-19 because of the accessibility of CoronaVac from China and local ChAdOx1 production capability [8]. To shorten the duration between prime and boosted injection. Moreover, there were reports of people having vaccine-induced adverse responses to the first shot of CoronaVac and this group was later boosted with the ChAdOx1 nCoV-19 vaccine, for this particular group, their immunogenicity was assessed and reported [9].

Based on immunogenicity data from patients who had AEs with CoronaVac and switched to ChAdOx1 in their second dosage. On 12 July 2021, the heterologous regimen was suggested to the national vaccination program with a 3–4 weeks interval between the first dose (CoronaVac) and the second dose (ChAdOx1 nCoV-19) [9]. However, even after the recommendation, the immunogenicity and adverse events (AEs) in a larger number of samples remained limited, and more data is urgently needed to support vaccination recommendations.

In this observational study, we aimed to compare the immunogenicity and safety of the CoronaVac-ChAdOx1 regimen with homologous CoronaVac, homologous ChAdOx1 regimen and the convalescent serum (CS) in COVID-19 patients (COVID-19 status confirmed positive by RT-PCR) at 4 weeks after diagnosis.

## 2. Method

### 2.1. Study Design

This pilot study was performed during May–August 2021; 354 individuals were recruited from the Bang Sue Vaccination Center, the Institute of Dermatology, the Ministry of Public Health in Thailand, and the Lak Si District from an observational study. The pilot study was recruited from two groups: (1) 120 people with natural COVID-19 infection and (2) 234 vaccinated people. The inclusion criteria for the vaccinated group were as follows: healthy individuals aged ≥ 18 years without a known history of COVID-19 or recent exposure to COVID-19 cases. We divided the vaccinated group into three groups based on vaccination type: 155 who received the CoronaVac-ChAdOx1 regimen, 32 who received homologous CoronaVac, and 47 who received homologous ChAdOx1. The participants with COVID-19 infection or exposure were verified by interviewing and re-checking with the national COVID-19 database. Participants in the CoronaVac-ChAdOx1 group (*n* = 155) had received CoronaVac followed by ChAdOx1 with 3–4-week intervals. The homologous CoronaVac group (*n* = 32) comprised participants who had received homologous CoronaVac vaccines after a 3–4-week interval. The homologous ChAdOx1 group (*n* = 47) comprised participants who had received homologous ChAdOx1 vaccines after a 10–12-week interval. The CS group (*n* = 120) refers to patients confirmed positive for COVID-19 by RT-PCR whose convalescent serum was collected at 4 weeks after diagnosis.

The study was conducted in accordance with the Declaration of Helsinki, The Belmont Report, CIOMS Guidelines, International Conference on Harmonization in Good Clinical Practice, and approved by the research ethics committee of the Department of Medical Sciences, Ministry of Public Health, Thailand (MOPH 0625/EC060, Date of approval 23 July 2021). Written informed consent was obtained from all participants. The diagram of participant enrollment in this study is illustrated in Figure 1. The CoronaVac-ChAdOx1 group (*n* = 155) was enrolled and classified into two subgroups (A, B) of participants. Group A included 30 participants who received CoronaVac and experienced at least one adverse event following immunization (AEFI) and were advised that they should be switched to second dose ChAdOx1. Within the first half of 2021, only ChAdOx1 was available in Thailand as an alternative to the inactivated viral vaccine. These patients were recruited from the Bang Sue Central Vaccination Center (BCVC), the largest center for COVID-19 vaccination in Thailand. Group B comprised 125 participants enrolled from the Ministry of Public Health vaccination center (MOPH-VC) at the permanent secretary office of the MOPH; at this center, those who were scheduled to receive a second dose of inactivated viral vaccine were offered to switch their second dose to ChAdOx1 nCoV-19, based on the national infectious disease academic subcommittee. The benefit of this CoronaVac-ChAdOx1 regimen, when compared with the homologous ChAdOx1 regimen, was an opportunity to achieve higher immunogenicity in shorter intervals. The benefits and risks of second vaccination with ChAdOx1 were also explained to participants at the enrollment.

Vaccination was performed according to the vaccination center guideline; the timing and lot of vaccines were recorded in the national vaccine information system, called the “Ministry of Public Health Vaccine Information Center (MOPH-VC)”. Immunogenicity analysis after the second vaccination at the Bang Sue Central Vaccination Center (BCVC) was offered as a test-based analysis voluntarily. All participants in the CoronaVac-ChAdOx1 group were invited for immunogenicity testing at 2 weeks and 4 weeks after the second vaccination with ChAdOx1 nCoV-19. The homologous ChAdOx1group (n = 47) participants from the BCVC and the homologous CoronaVac group (n = 32) participants from the MOPH-VC were invited for immunogenicity testing at 4 weeks after the second vaccination dose. All participants provided written informed consent to have their immunogenicity and adverse events included in this study.

### 2.2. Quantification of SARS-CoV-2 Anti-SRBD Antibody

The level of immunoglobulin class G (IgG) antibodies against the receptor-binding domain (RBD) of the S1 subunit spike protein of SARS-CoV-2 was measured and quantified in human serum or plasma using the ARCHITECT System (Abbott, Abbott Park, IL, USA) chemiluminescent microparticle immunoassay (CMIA) (SARS-CoV-2 IgG II Quant, Abbott Ireland, Sligo, Ireland), measuring a reportable range from 6.8 to 80,000.0 Abbott Arbitrary Unit (AU/mL) (up to 40,000 AU/mL with onboard 1:2 dilution). Values higher than 50 AU/mL were considered positive. Based on the evaluated dilutions of the World Health Organization (WHO) International Standard (NIBSC Code 20-136) for anti-SARS-CoV-2 human immunoglobulin in WHO binding antibody unit (WHO BAU/mL) with the SARS-CoV-2 IgG II Quant assay and internal reference calibrators, the correlation between the AU/mL unit and the WHO unit (BAU/mL unit is at 0.142 × AU/mL) had a correlation coefficient of 0.999.

### 2.3. Pseudotype-Based Microneutralization Assay against SARS-CoV-2

The pseudotype-based microneutralization assay was implemented at the Virology and Cell Technology Laboratory, National Center for Genetic Engineering and Biotechnology (BIOTEC) by the following protocol. Pseudotyped viruses (PVs) were produced in HEK293T/17 cells. HEK293T/17 producer cells were sub-cultured in 6-well plates (Thermo Fisher Scientific, Waltham, MA, USA) and co-transfected at 80–90% confluence with the p8.91 lentiviral packaging plasmid (500 ng), the pCSFLW firefly luciferase reporter plasmid (1 µg), and the pCAGGS plasmid encoding codon-optimized SARS-CoV-2 spike (1.5 µg) [10].The transfection was performed using polyethylenimine (PEI). The cell supernatants were collected 72 h before storage in microcentrifuge tubes at −80 °C.

Pseudotype-based microneutralization (pMN) assays were performed as previously described [10] with some modifications. Briefly, human sera from SARS-CoV-2 seropositive and seronegative patients were subjected to two-fold dilution (starting from 1:40) in a white, flat-bottomed 96-well tissue culture-treated plate, and 50 µL of PV containing supernatant diluted in complete DMEM to give 5 × 10^5^ RLU equivalent was added per well. The plate was then centrifuged at 300× *g* for 3 min before incubation in a humidified cell culture incubator for 1 h at 37 °C and 5% CO_2_. Subsequently, HEK293T cells stably expressing human ACE2 and TMPRSS2 (1.5 × 10^4^ cells/well) were mixed with the PV–serum complex before incubation in a humidified cell culture incubator for 48 h at 37 °C and 5% CO_2_. To measure luminescence activity, equal volumes of Bright-Glo™ reagent (Promega, Madison, WI, USA) and phosphate-buffered saline (PBS) were mixed and 25 µL was added to each well; luminescence output was measured using a Synergy™ HTX Multi-Mode Microplate Reader (BioTek, Winooski, VT, USA) after 5 min incubation at room temperature. Analysis was performed by non-linear regression after normalization to 100 and 0% neutralization using GraphPad Prism Software.

### 2.4. Plaque Reduction Neutralization Test (PRNT)

The PRNT in this study was developed and tested by the Institute of Biological Products, a WHO-contracted laboratory at the Department of Medical Sciences. Vero cells were seeded at 2 × 10^5^ cells/well/3 mL and placed in an incubator at 37 °C and 5% CO_2_ for 1 day. Test serum was initially diluted at 1:10, 1:40, 1:160, and 1:640. The SARS-CoV-2 virus was diluted in a culture medium to yield 40–120 plaques/well in the virus control wells. Cell control wells, convalescent patient serum, and normal human serum were also included as assay controls. The neutralization was performed by mixing an equal volume of diluted serum and the optimal plaque numbers of SARS-CoV-2 virus at 37 °C in a water bath for 1 h. After removing the culture medium from Vero cell culture plates, 200 µL of the virus–serum antibody mixture was inoculated into monolayer cells, and then the culture plates were rocked every 15 min for 1 h. Three milliliters of overlay semisolid medium (containing 1% carboxymethylcellulose (Sigma Aldrich, St. Louis, MO, USA), with 1% of 10,000 units/mL penicillin–10,000 µg/mL streptomycin (Sigma, St. Louis, MO, USA) and 10% FBS) was replaced after removing excessive viruses. All plates were incubated at 37 °C and 5% CO_2_ for 7 days. Cells were fixed with 10% (*v*/*v*) formaldehyde and then stained with 0.5% crystal violet in PBS. The number of plaques formed was counted in three wells, and the percentage of plaque reduction at 50% (PRNT_50_) was calculated. The PRNT_50_ titer of the test sample was defined as the reciprocal of the highest test serum dilution for which the virus infectivity was reduced by 50% when compared with the average plaque counts of the virus control and was calculated by using a four-point linear regression method. Plaque counts for all serial dilutions of serum were scored to ensure that there was a dose response.

### 2.5. Monitoring of Adverse Events after Vaccination

Adverse events (AEs) were determined according to the SOP 43-05-17-CL-006 Handling and Reporting of Adverse Events protocol and were retrieved from questionnaires and/or telephone-based interviews. The registered nurse contacted all the subjects, and none of them reported any serious adverse effects. The rates of each adverse drug reaction were reported. The Common Terminology Criteria for Adverse Events—Version 5.0 were used to evaluate the severity of AEs.

### 2.6. Statistical Analysis

All statistical analyses were performed using R (version 4.0.2) [11] and RStudio (version 1.3.1093) [12]. The anti-SARS-CoV-2 RBD IgG level, PRNT_50_ titer, and PVNT_50_ titer were presented as geometric mean titers (GMTs) with 95% confidence intervals (CI). The differences in the antibody levels in the CoronaVac-ChAdOx1, homologous CoronaVac, homologous ChAdOx1, and CS groups at 4 weeks and in the CoronaVac-ChAdOx1 group between 2 and 4 weeks were tested using the non-parametric Mann–Whitney U test [13] (available within the R package rstatix). The Wilcoxon signed-rank test, a non-parametric statistical hypothesis test for paired data [13], was applied to compare the PRNT_50_ titers and PVNT_50_ titers among selected SARS-CoV-2 variants. Spearman’s rank correlation coefficient11 (R function cor. test) was used to assess the relationship between SARS-CoV-2 variants and the neutralization assays (PVNT_50_ and PRNT_50_). A *p*-value < 0.05 was considered significant. The AEs outcomes were provided in the form of rates and the chi-square test was used to determine the significant difference between two groups.

## 3. Results

From May to August 2021, 354 participants were included in this pilot study. The demographic data of the vaccination groups are described in Table 1.

### 3.1. Immunogenicity Profiles

#### 3.1.1. Anti-SRBD Levels of CoronaVac-ChAdOx1, Homologous CoronaVac, Homologous ChAdOx1, and CS Groups

At 2 weeks, 149 of the 155 participants completed their visits in the CoronaVac-ChAdOx1 group, and six participants missed their 2-week visit but returned for the visit at 4 weeks; the geometric mean titer (GMT) of anti-SRBD levels was 873.9 BAU/mL (95% CI 768.4–993.8) (Figure 2).

At 4 weeks, 137 of the 155 participants remained in the CoronaVac-ChAdOx1 group, with 18 participants excluded. From these participants, one was infected with HIV with a low CD4 count, two were infected with COVID-19 after the visit at 2 weeks, and 15 were lost to follow-up; the GMT of anti-SRBD levels was 639 BAU/mL (95% CI 563–726).

The GMTs of anti-SRBD levels in the CS group, the homologous ChAdOx1 group, and the homologous CoronaVac group was 227.9 (95% CI 172–303), 211.1 (95% CI 161–277), and 108.2 BAU/mL (95% CI 77–152), respectively.

The GMTs of anti-SRBD levels of the CoronaVac-ChAdOx1 group at 2 weeks was significantly higher than that at 4 weeks (*p* = 0.00114) (Figure 2). The GMT of anti-SRBD levels of the CoronaVac-ChAdOx1 group at 4 weeks was significantly higher than those of the homologous CoronaVac group, homologous ChAdOx1 group, and CS group (*p* < 0.001) (Figure 3).

#### 3.1.2. Plaque Reduction Neutralization Antibody Test on Variants of Concern (PRNT)

Sera collected from 19 of the 30 participants in group A of the CoronaVac-ChAdOx1 regimen following the second vaccination dose after 2 weeks were analyzed for neutralizing antibodies (NAbs) by the PRNT_50_ method against four variants of SARS-CoV-2 (Wuhan strain or wild-type, Alpha variant, Delta variant, and Beta variant).

The GMT of PRNT_50_ was the highest against the wild-type (434.5 BAU/mL, 95% CI 326–579), being significantly higher compared to that against the Alpha variant (80.4 BAU/mL, 95% CI 56–115), Delta variant (67.4 BAU/mL, 95% CI 48–95), and Beta variant (19.8 BAU/mL, 95% CI 14–30) (*p* < 0.001) (Figure 4). Moreover, the PRNT_50_ level showed a correlation with the anti-SRBD level with the wild-type (ρ = 0.82, *p* < 0.001), Alpha variant (ρ = 0.66, *p* = 0.0027), Beta variant (ρ = 0.074, *p* = 0.76), and Delta variant (ρ = 0.17, *p* = 0.48) (Figure 5).

#### 3.1.3. Pseudo Neutralization Antibody Test for Variants of Concern (PVNT)

Serum from 30 participants in group A of the CoronaVac-ChAdOx1 regimen were collected at 2 weeks after second dose, were analyzed for neutralizing antibody (NAb) using the PVNT_50_ method against four variants of SARS-CoV-2 (Wuhan strain or wild-type, Alpha variant, Beta variant, and Delta variant). The GMT of PVNT_50_ was the highest against the wild-type (432.1, 95% CI 299–624), being significantly greater than those for the Delta (178.3 BAU/mL, 95% CI 107–298) (*p* = 0.029), Alpha (163.9 BAU/mL, 95% CI 89–301) (*p* = 0.029), and Beta variants (42.2 BAU/mL, 95% CI 10–173) (*p* = 0.002) (Figure 6). Furthermore, the PVNT_50_ level had a correlation with the anti-SRBD level in the wild-type (ρ = 0.39, *p* = 0.16), Alpha variant (ρ = 0.47, *p* = 0.076), and Delta variant (ρ = 0.26, *p* = 0.35) (Figure 7).

### 3.2. Adverse Events Following Immunization in Three Vaccination Regimens

The most common systemic AEs after the second vaccination dose within 4 weeks were feeling feverish (67.1%) in the CoronaVac-ChAdOx1 group and feeling feverish (23%) in the homologous ChAdOx1 group and headache (19%) in the homologous CoronaVac group (Appendix A). The most common local AE was injection site pain in all three groups (35.06%, 25%, and 9%, respectively). The severity of most of the local and systemic AEs was mild or moderate and improved within a few days. There were no serious AEs reported.

## 4. Discussion

In our pilot study, the GMT of anti-SRBD in the CoronaVac-ChAdOx1 vaccination group was significantly higher than in the homologous ChAdOx1, homologous CoronaVac, and CS groups. This is comparable to previous data on heterologous prime–boost techniques [14].The higher level of antibody response and neutralizing activity against the Delta variant in the CoronaVac-ChAdOx1 group is encouraging for the national vaccine program implementation.

Interestingly, the higher anti-SRBD IgG level in the CoronaVac-ChAdOx1 group compared to the homologous ChAdOx1 and homologous CoronaVac groups is in concordance with a previous smaller study [15]. The study [15] shows that participants who received an initial dose of CoronaVac followed by ChAdOx1 had higher antibody levels than those who received the homologous inactivated vaccine (CoronaVac—BBIBP-CorV) and homologous ChAdOx1; it also agrees with the research [16] carried out in showing that participants who received CoronaVac-ChAdOx1 vaccination had significantly higher antibody levels than those who received homologous CoronaVac regimen.

Additionally, this result could be explained with further support from a biological study [17] in which the author demonstrated in mice that a first dose of inactivated viral vaccine followed by adenovirus vector vaccine administration induced a significantly more prominent T cell response than two doses of the inactivated vaccine. This might help increase B cell differentiation into plasma B cells that could provide a better antibody response to specific antigens compared to two doses of inactivated vaccines [18]. Furthermore, reducing anti-vector and anti-non-spike protein immune activation might be a plausible mechanism explaining this immunogenicity, as heterologous prime–boost vaccination resulted in increased anti-spike protein immune responses [19]. Increases in anti-spike IgG, anti-RBD IgG, and neutralizing antibody titers were related to reduced disease severity in symptomatic COVID-19 infection [20].

In addition, the antibody levels of the homologous CoronaVac group at 4 weeks were lower than those of the CS group. This agrees with a prior study from 2021 showing that the S1-RBD-binding IgG titer of natural infection was higher than that from homologous CoronaVac vaccination [21].

The most important finding is that the CoronaVac-ChAdOx1 regimen provided a higher Nab level at 2 weeks against the Alpha variant than the Beta variant. This is compatible with a previous study utilizing a pseudovirus neutralization assay, in which serum from CoronaVac vaccinees collected 14 days following the second dose of the vaccine presented higher Beta resistance to neutralization than Alpha [22].Moreover, this Nab level at 2 weeks against all VOCs decreased significantly compared to wild-type [21].

The Nab level was the highest for the wild-type variant, followed by the Alpha variant and Delta variant, with the lowest Nab level for the Beta variant. This is consistent with a study that analyzed neutralization capacity by protein-based 206 ACE2 RBD competition assay, finding it to be highest in the wild-type, followed by the Alpha, Delta, and Beta variants [19]. In addition, various mutations in the SARS-CoV-2 spike protein’s N-terminal domain (NTD) and receptor-binding domain (RBD) may enhance the immune evasion capability of these variations [23]

Participants in the CoronaVac-ChAdOx1 schedule showed high PRNT_50_ and PVNT_50_ titers against the Delta variant. Immune protection against SARS-CoV-2 infection symptoms is highly correlated with levels of neutralizing antibodies [20].

This evidence provides important supporting information for this CoronaVac-ChAdOx1 vaccination schedule in Thailand. The availability of immunogenicity data from the CoronaVac-ChAdOx1 vaccination schedule is vital for the consideration of the national vaccination program to determine the best vaccine schedule that could stimulate a higher level of immunogenicity. As immunogenicity studies may not be feasible for every country, low- and middle-income countries where inactivated viral vaccines and viral vector vaccines are available may use this evidence to support their decision to use alternative vaccination schedules.

While multiple inactivated viral vaccines and viral vector vaccines are available, not all combinations of vaccine regimens produce satisfactory immunogenicity responses. Early analysis in Thailand suggested that viral vector vaccine administration followed by inactivated viral vaccine administration induced unsatisfactory immunogenicity [24]. While a correlation of protection for COVID-19 vaccines has not been officially established by the World Health Organization, neutralizing activity to SARS-CoV-2 variants is likely related to the effectiveness of the vaccination schedule [23]. Each country can measure the immunogenicity of these vaccination schedules in their target population and utilize this information to guide the vaccination schedules at the national level. In countries where CoronaVac and ChAdOx1 nCoV-19 are available, such as Thailand, using this heterologous vaccination schedule is likely to enable the highest level of immunogenicity to the Delta variant.

The introduction of the heterologous administration of ChAdOx1 nCoV-19 followed by mRNA vaccination was evaluated in terms of the vaccination effectiveness (VE) of this schedule in fully vaccinated individuals, and the real-world VE of this vaccination schedule was 88% (95% CI, 83–92%) [25].A large-scale evaluation of the safety and effectiveness of the IV-AV regimen is still ongoing in Thailand.

The AEs of the CoronaVac-ChAdOx1 group within 4 weeks after the second vaccination dose were well tolerated and generally safe. The systemic and local AEs rates of the heterologous regimen were higher compared to the homologous CoronaVac and homologous ChAdOx1 regimens, but the severity of the AEs was mild to moderate, without any report of serious AEs. The rates of local and systemic AEs in CoronaVac-ChAdOx1 regimens are higher than homologous regimens but have no serious adverse events [15,16]. The findings in this study are consistent with two earlier studies in Thailand [15,16]. Moreover, the World Health Organization (WHO) launched an interim recommendation for heterologous COVID-19 vaccination at the end of 2021 [26]. The most common local AEs of the homologous CoronaVac group were comparable to previous reports [27]. A recent study reported a higher rate of systemic and local vaccine reactions after heterologous CoronaVac-ChAdOx1 vaccination compared with homologous ChAdOx1 and CoronaVac vaccination [15,16]. The COM-COV trial revealed that participants receiving heterologous regimens containing the ChAdOx1 nCoV-19 and BNT162b2 vaccines showed increased reactogenicity compared to those receiving homologous regimens [28]. Multiple similarities in study design (observational study), vaccine interval, and baseline characteristics of participants can be described in this correspondence.

The question that remains to be answered concerns the duration of immunogenicity, which may need follow-up studies at 3 months, 6 months, 9 months, and 1 year. The decreasing antibody levels from 2 to 4 weeks suggested that the immunogenicity level may last from 3 to 6 months, and the temporal relationship of effectiveness should be monitored carefully by the national vaccination program. A booster dose at 3 to 6 months is likely necessary for boosting immunogenicity after the CoronaVac-ChAdOx1 vaccination, and the type and timing of the vaccine should be determined in a follow-up study from this cohort of vaccine recipients.

There are several limitations to our research. First, our study did not collect comparison data on immunity levels at baseline and after the first vaccine dose; hence, we could not exclude the effect of the previous infection on immunogenicity. The COVID-19 infection rate in Thailand was generally unremarkable before July 2021; however, there is a small possibility of having included people previously infected with COVID-19. Nevertheless, these individuals should not affect the immunogenicity results in this study as the immunity induced by infection was much higher in previously infected individuals compared with single-dose vaccinated individuals.

A larger sample size is needed to detect uncommon adverse events, such as vaccine-induced thrombotic thrombocytopenia, and further studies should be conducted at the time of implementation of this vaccine schedule in the national vaccine program. Moreover, the PRNT_50_ and PVNT_50_ were assessed only in the CoronaVac-ChAdOx1 regimen group, thus limiting the comparability of the CoronaVac-ChAdOx1 regimen with other vaccine schedules. Additionally, the limited time for this study resulted in a smaller sample size that had neutralizing activities; however, the high level of neutralizing antibodies suggested that the sample size is sufficient for concluding that the CoronaVac-ChAdOx1 schedule is significantly better at inducing humoral immunity than the homologous CoronaVac schedule and is comparable to the homologous c schedule.

The pilot study’s limitations include the inadequacies on the duration of immunogenicity effects and effectiveness against variants that were not described in this study. Several studies demonstrated that both the homologous and heterologous vaccination regimens using wild type SARS-CoV-2 seem to have lower efficacy against the Delta and Omicron variants [29,30]. Recently, some studies on the CoronaVac-ChAdOx1 regimen in Thailand had indicated that the duration of protection is between 16 and 20 weeks after receiving the second dose. Therefore, we suggest that individuals who received heterologous vaccination may consider the booster dose at 16–20 weeks after the second dose [15]. We estimated the duration of immunogenicity of the homologous ChAdOx1 regimen to be around 6 months based on a recent study in Finland [30]. According to other studies, breakthrough infection on the homologous CoronaVac regimen was detected at approximately 88 days (IQR 68–100) after the second dosage [31]. As a result, Thailand recommends that people receive a booster dose after the second dosage at three months for homologous CoronaVac and six months for homologous ChAdOx1.

Finally, our pilot study offers crucial real-world proof of the safety and immunogenicity of heterologous CoronaVac–ChAdOx1. The CoronaVac-ChAdOx1 vaccination is a mixed regimen that induced higher immunogenicity with a shorter duration to peak immunogenicity compared to the homologous CoronaVac schedule. In a situation where the viral vector vaccine is inadequate, we should consider this vaccine schedule for responding to the Delta variant. This initial assessment encourages future research of heterologous prime–boost vaccination regimens for COVID-19.

## Figures and Tables

**Figure 1 vaccines-10-00536-f001:**
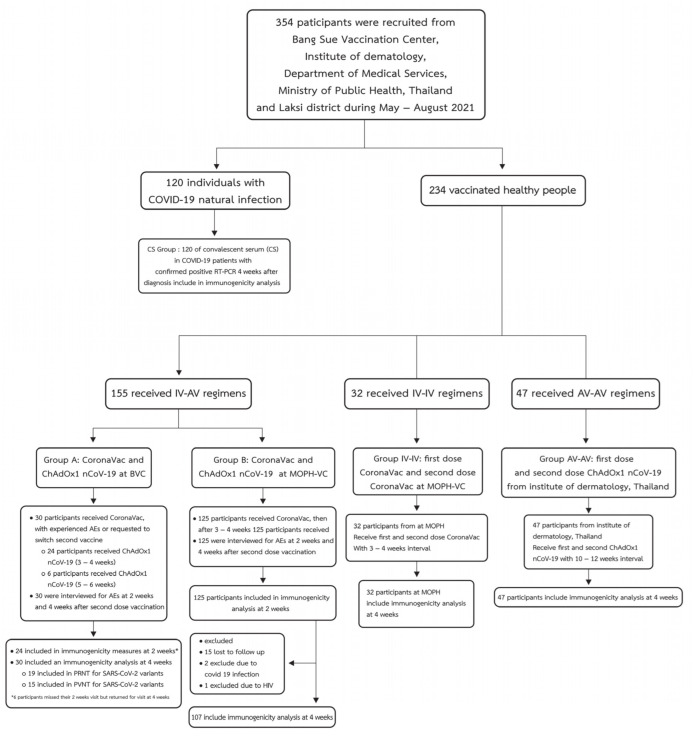
Enrollment diagram of the study population. In total, 354 participants were recruited from Bang Sue Central Vaccination Center, Institute of Dermatology, Ministry of Public Health, Thailand, and Lak Si District. The cohort was divided into 2 groups: (1) 120 individuals with natural COVID-19 infection and (2) 234 vaccinated people. The vaccinated group was classified into 3 subgroups according to the type of vaccine: 155 IV-AV, 32 IV-IV, and 47 AV-AV courses. (IV stands for CoronaVac, AV stands for ChAdOx1, CS group stands for convalescent serum in COVID-19 patients with confirmed positive RT-PCR 4 weeks after diagnosis).

**Figure 2 vaccines-10-00536-f002:**
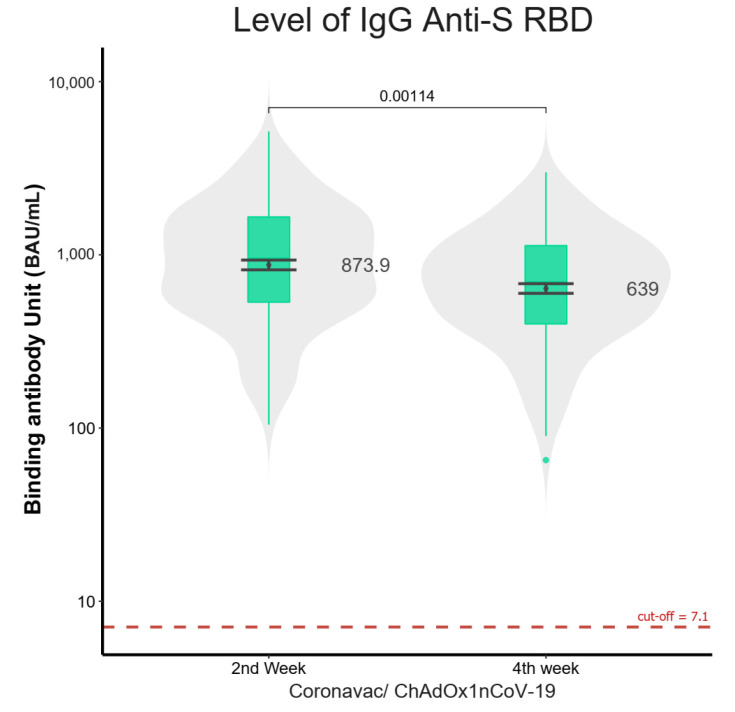
Violin plots comparing level of IgG anti-SRBD titer in CoronaVac-ChAdOx1 group at 2 and 4 weeks. Boxplots show geometric mean and IQRs. Level of IgG anti-SRBD titer at 2 weeks was significantly (*p* = 0.00114) (873.9 BAU/mL, 95% CI 768–994) higher than at 4 weeks (639 BAU/mL, 95% CI 563–726). Analysis was performed using the non-parametric Mann–Whitney U test.

**Figure 3 vaccines-10-00536-f003:**
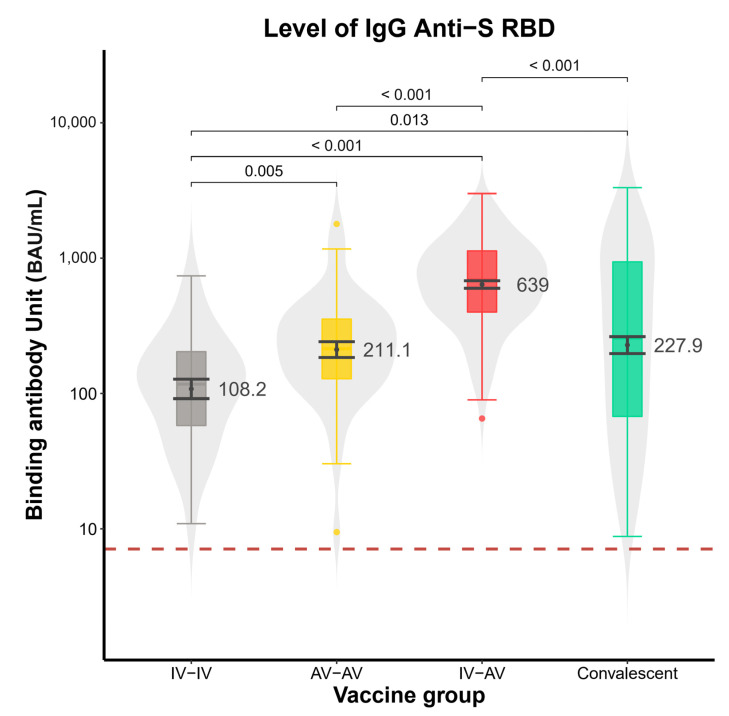
Violin plots of level of IgG anti-SRBD titer in 4 groups. Boxplots show geometric mean and IQRs. Serum was collected at 4 weeks. IV-AV group showed a significantly higher level (*p* < 0.0001) (639 BAU/mL, 95% CI 563–726) than the AV-AV (211.1 BAU/mL, 95% CI 161–277), Convalescent (227.9 BAU/mL, 95% CI 172–303) (*p* < 0.0001), and IV-IV groups (108.2 BAU/mL, 95% CI 77–152) (*p* < 0.0001). Analysis was performed using the non-parametric Mann–Whitney U test. (IV stands for CoronaVac, AV stands for ChAdOx1) Abbreviations: IV-IV = CoronaVac/CoronaVac; AV-AV = ChAdOx1 nCoV-19/ChAdOx1 nCoV-19; IV-AV = CoronaVac/ChAdOx1 nCoV-19.

**Figure 4 vaccines-10-00536-f004:**
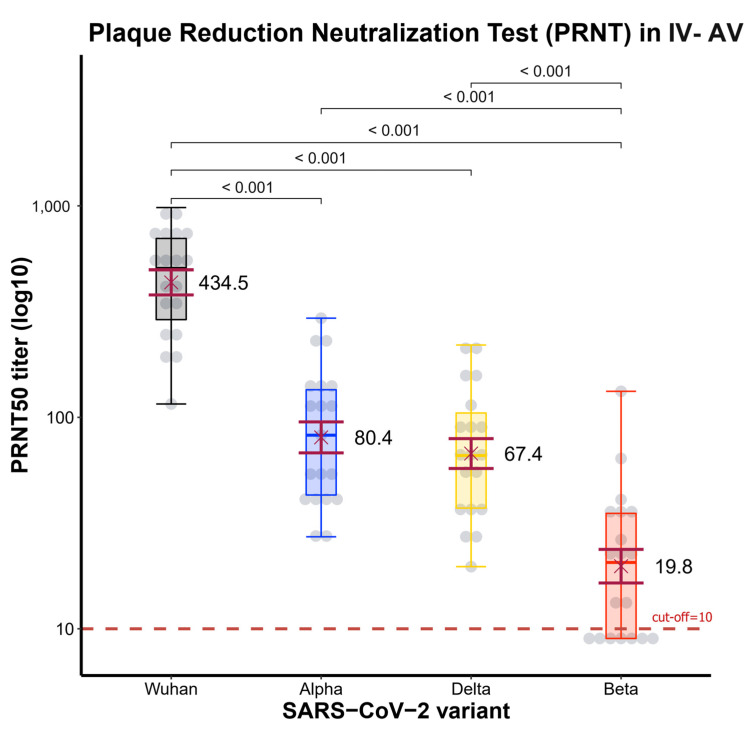
Boxplots of PRNT_50_ level in CoronaVac-ChAdOx1 group. Boxplots indicate geometric mean PRNT_50_ titer to virus variants and IQRs. Serum was collected 2 weeks after participants in the CoronaVac-ChAdOx1 group received the vaccination. The GMT of PRNT_50_ was highest against the wild-type (434.5 BAU/mL, 95% CI 326–579)—significantly greater than against the Alpha variant (80.4 BAU/mL, 95% CI 56–115), Delta variant (67.4 BAU/mL, 95% CI 48–95) (all *p* < 0.001), and Beta variant (19.8 BAU/mL, 95% CI 14–30) (*p* < 0.001). The horizontal dotted line indicates the positive detection (10 units). The Wilcoxon signed-rank test was used for assessment.

**Figure 5 vaccines-10-00536-f005:**
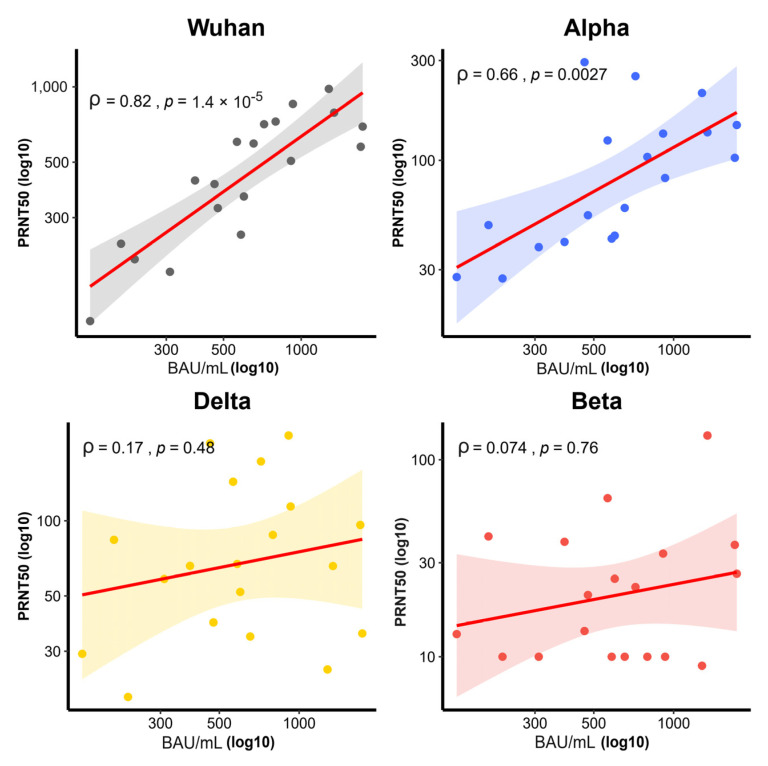
Correlation between PRNT_50_ titer with Ab to anti-SRBD level by virus variant group. Correlation (solid line), 95% confidence intervals (curved line). Wild-type (ρ = 0.82, *p* < 0.001), Alpha variant (ρ = 0.66, *p* = 0.0027), Beta variant (ρ = 0.074, *p* = 0.76), and Delta variant (ρ = 0.17, *p* = 0.48). Spearman’s rank correlation coefficient was used for assessment. Serum was collected 2 weeks after participants in the IV-AV group received a vaccination.

**Figure 6 vaccines-10-00536-f006:**
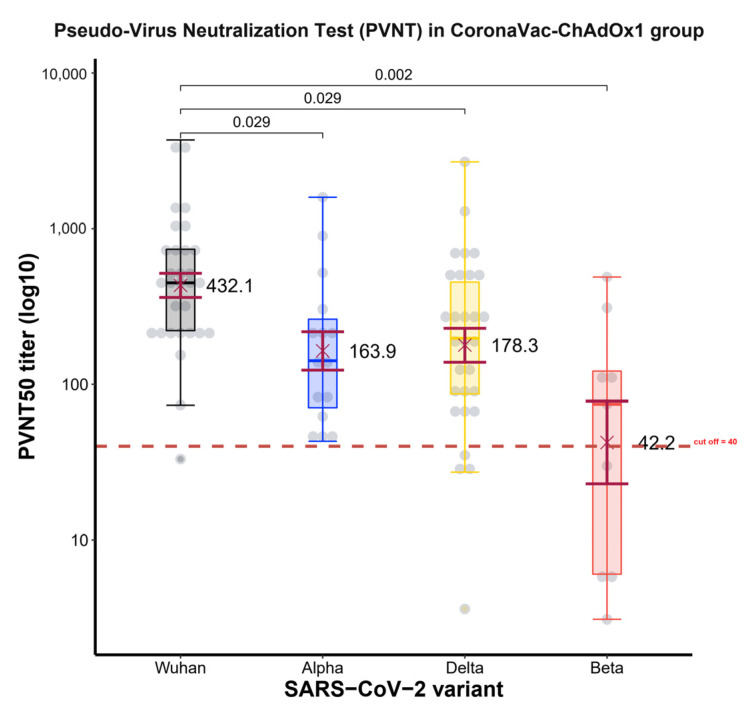
Boxplots of PVNT_50_ level in CoronaVac-ChAdOx1 group. Boxplots indicate geometric mean PVNT_50_ titer to virus variant and IQRs. Serum collection was performed 2 weeks after participants in the CoronaVac-ChAdOx1 group received the vaccination. The GMT of PVNT_50_ was highest against the wild-type (432.1 BAU/mL, 95% CI 299–624), being significantly greater than against the Delta (178.3 BAU/mL, 95% CI 107–298) (*p* = 0.029), Alpha (163.9 BAU/mL, 95% CI 89–301) (*p* = 0.029), and Beta variants (42.2 BAU/mL, 95% CI 10–173) (*p* = 0.002). The horizontal dotted line indicates the positive detection (40 units). The Wilcoxon signed-rank test was used for analysis.

**Figure 7 vaccines-10-00536-f007:**
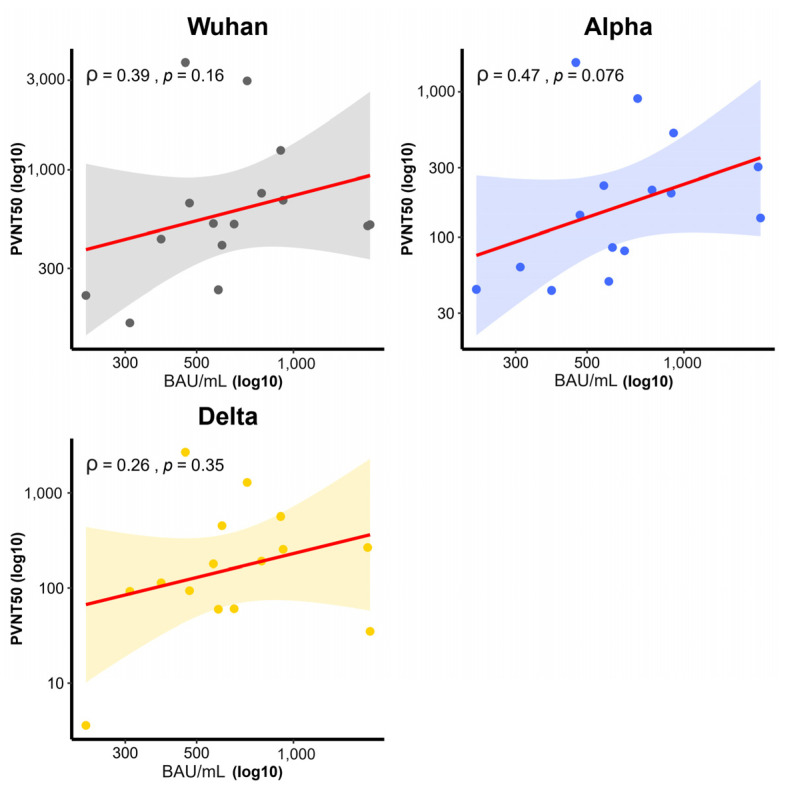
Correlation between PVNT_50_ titer with Ab and anti-SRBD level by virus variant group. (Wild-type, Alpha variant, and Delta variant) Correlation (solid line), 95% confidence intervals (curved line).Wild-type (ρ = 0.39, *p* = 0.16), Alpha variant (ρ = 0.47, *p* = 0.076), and Delta variant (ρ = 0.26, *p* = 0.35). Spearman’s rank correlation coefficient was used to assess the correlation. Serum collection was conducted 2 weeks after participants in the CoronaVac-ChAdOx1 group received a vaccination.

**Table 1 vaccines-10-00536-t001:** Demographic data of the study population.

	GROUP A ^1^ CoronaVac/ChAdOx1	GROUP B ^2^ CoronaVac/ChAdOx1	CoronaVac/CoronaVac	ChAdOx1/ChAdOx1	CS ^3^
SEX	MOPH-VC	BCVC	MOPH-VC	ISDT ^4^	Lak Si District
male (%)	61 (49%)	15 (50%)	6 (18.75%)	7 (14.9%)	71 (59%)
Female (%)	64 (51%)	15 (50%)	26 (81.25%)	40 (85.1%)	49 (41%)
Total	125	30	32	47	120
Age group					
18–49	99	25	24	28	91
male	45	13	2	3	51
Female	54	12	22	25	40
50–70	26	5	8	19	29
male	16	2	4	4	20
Female	10	3	4	15	9
median age group ± sd	40 ± 8.8	40 ± 9.24	45 ± 8.8	53.5 ± 14.6	43 ± 8.9
History of COVID-19 Infection	0	0	0	0	120

^1^ 125 participants at MOPH-VC, ^2^ 30 participant at BCVC,  ^3^ convalescent serum from participants from Lak Si district with positive RT-PCR COVID-19 after 4 weeks, ^4^ Institute of Dermatolog, Ministry of Public Health, Bangkok.

## Data Availability

The data that supports the findings of this study are available from the corresponding author upon reasonable request.

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
