# Peer review of "The Pilot Study of Immunogenicity and Adverse Events of a COVID-19 Vaccine Regimen: Priming with Inactivated Whole SARS-CoV-2 Vaccine (CoronaVac) and Boosting with the Adenoviral Vector (ChAdOx1 nCoV-19) Vaccine"

_vaccines, 2022, doi:10.3390/vaccines10040536_

Round 1

Reviewer 1 Report

  1. Authors conducted an observational study to compare the immunogenicity and adverse events of a SARS-CoV-2 vaccination regime based on Coronavac and AstraZeneca shots as a part of a national strategy in Thailand. They concluded that this approach seems safe and efficacious. 
  2. English needs extensive copyediting by a Native speaker. 
  3. Main concern: statistical representativeness of the inactivated/inactivated and vector/vector groups, in particular regarding male participants. 
  4. Abstract: the aim of the study states "to compare the immunogenicity and adverse events of three regimens (IV-AV, IV-IV, and AV-AV) and immunogenicity of convalescent serum (CS) in COVID-19 patients with confirmed positive RT-PCR 4 wks after di-25 agnosis (CS group)". However, the study design included also a control group (234 healthy individuals). Please, clarify.
  5. Abstract: define only the acronyms that are used in the abstract. 
  6. Use consistently COVID-19, covid-19, covid, COVID19, SARS-CoV-2 infection, etc across the text. 
  7. Introduction: Please, break it down to at least two more paragraphs and try and shorten it to the essential information to grasp the justification and aims of the study. 
  8. Ethics: provide further information about the IRB institution. 
  9. Figure 1: include a title. Clarify numbers so they can be easily tracked back to the figures provided at the stem of the design (ie, 120 patients and 234 controls). I'd suggest to merge Fig 1 and 2 in a unique flowchart.
  10. Figure 2 also lacks a title. 
  11. Results: From April to August 2021 or rather May to August as stated in the abstract?
  12. Figure 7 lacks a title. 
  13. Discussion: please, mention whether the sample size do limit, beside AE conclusions, conclusions overall. Given that the Thailand national strategy mostly consisted on the IV-AV regimen from a certain point in time onwards, the cohort could have been easily quite larger.
  14. Discussion: would you care commenting on the duration of immunogenicity effects in the different groups?
  15. Discussion: could you speculate and mention recent studies on the efficacy of the assessed vaccines and the combined regimen on non-Delta variants?

Author Response

Response to Reviewer 1 Comments

Point 1: Main concern: statistical representativeness of the inactivated/inactivated and vector/vector groups, in particular regarding male particiants

Response 1: Regarding this concern, we compared the anti-S RBD qualitative level (BAU/mL) between males and females using the Wilcoxon rank sum test on R software. There was no statistically significant difference between males and females in either the inactivated/inactivated group, homologous CoronaVac (p-value = 0.436) or vector/vector group, homologous ChAdOx1 (p-value = 0.209). (This additional analysis is prepared into supplementary figure 1 and supplementary table 1)

Point 2: Abstract: the aim of the study states "to compare the immunogenicity and adverse events of three regimens (IV-AV, IV-IV, and AV-AV) and immunogenicity of convalescent serum (CS) in COVID-19 patients with confirmed positive RT-PCR 4 wks after di-25 agnosis (CS group)". However, the study design included also a control group (234 healthy individuals). Please, clarify.

Response 2: The cohort was recruited from 2 groups: (1) 120 people with natural COVID-19 infection from Lak Si District during May 2021 and (2) 234 vaccinated healthy individuals from two vaccination centers a) Bang Sue Vaccination Center, Institute of Dermatology and b) Office of Permanent Secretary Vaccination Center, Ministry of Public Health. We enrolled 3 groups according to the vaccine regimens: 155 obtaining CoronaVac- ChAdOx1 regimens, 32 homologous CoronaVac, and 47 homologous ChAdOx1 regimens.

Point 3: Abstract: define only the acronyms that are used in the abstract.

Response 3:

1.IV stands for inactivated whole virus vaccine (CoronaVac)

  1. AV stands for adenoviral vector vaccine (ChAdOx1 nCoV-19, AstraZeneca)

3.CS group stands for convalescent serum in COVID-19 patients with confirmed positive RT-PCR 4 wks after diagnosis

  1. anti-S RBD stands for antibodies against the receptor-binding domain of the SARS-CoV-2 spike protein S1 subunit
  2. NAbs stands for neutralizing antibody
  3. PRNT stands for plaque reduction neutralization test
  4. PVNT stands for pseudotype-based microneutralization test against
  5. VOCs stands for SARS-CoV-2 variants of concern.

9.AEs stands for adverse events

Point 4: Introduction: Please, break it down to at least two more paragraphs and try and shorten it to the essential information to grasp the justification and aims of the study. 

Response 4 This is the edited text in the introduction:

Due to the limited COVID-19 vaccine availability in early 2021, low- and middle-income countries relied on the accessible inactivated whole-virus vaccine CoronaVac1 (Sinovac Biotech, Beijing, China) and viral vector vaccines such as the ChAdOx1 nCoV-19 vaccine2. The heterologous prime–boost vaccination regimen induces better immunogenicity for various vaccines3 and is regularly practiced in routine vaccination programs, such as the annual influenza vaccination program, and in vaccination programs for other diseases where vaccines from various manufacturing platforms are usually considered replaceable.

The Delta variant outbreak from the beginning of 2021 in India was alarmingly devastating, with an estimated number of deaths 10 times higher than the reported 400,000 deaths due to its higher transmissibility, enhanced severity, and vaccine escape capability3. Based on these characteristics, the Delta variant was recognized as a variant of concern (VOC) by Public Health England, the US Center for Disease Control, and the World Health Organization (WHO )4. In mid-2021, the WHO issued global notices of the Delta variant as the dominant variant and notified vaccination programs to accelerate the implementation of any vaccine that is readily accessible.

The Delta variant in Thailand was first detected in May 2021 and became Thailand’s dominant variant of COVID-19 in the first week of August 20215 . Consequently, Thailand began a COVID-19 vaccination program with CoronaVac and ChAdOx1 nCoV-19, with diplomatic supported accessibility to the inactivated viral vaccine from China and the local contracted manufacturing capacity of AV6. When the vaccination program was initially implemented from March to May 2021, a group of individuals who experienced vaccine-induced adverse reactions from CoronaVac were offered the ChAdOx1 nCoV-19 vaccine as the only possible alternative to CoronaVac, and their immunogenicity was measured and reported4.

Based on immunogenicity data from those who experienced AEs from CoronaVac, a heterologous vaccine regimen involving priming with an inactivated SARS-CoV-2 vaccine followed by boosting with an adenoviral vector vaccine[MH4]  (ChAdOx1 nCoV-19) was proposed and recommended based on the neutralizing antibody data from this vaccine regimen by the national vaccination program on 12th July 2021.

Thailand proposed a heterologous prime–boost regimen with a 21-day interval between the first dose (CoronaVac) and the second dose (ChAdOx1 nCoV-19).However, after the recommendation, the immunogenicity in a larger number of samples and detailed information of common reactogenicity, especially adverse reactions to the second vaccination dose of ChAdOx1 nCoV-19 after the inactivated viral vaccine, were still limited and are urgently needed for informing vaccination recommendations.

In this observational study, we aimed to compare the immunogenicity and safety of the CoronaVac- ChAdOx1 regimen, homologous CoronaVac regimen, and homologous ChAdOx1regimen and the immunogenicity of convalescent serum (CS) in COVID-19 patients (COVID-19 status confirmed positive by RT-PCR). The CS group was followed until 4 weeks after diagnosis.

Point 5 Ethics: provide further information about the IRB institution.

Response 5 in line 510-513  in Ethics statement section

The ethical approval of this study was approved by the ethical committee of the Department of Medical Sciences with approval number; MOPH 0625/EC060

Contact : 88/7 Moo 4, Department of Medical Sciences, Building 14, 7th Floor, Room 720 ,Tiwanon Road, Talat Khwan Subdistrict, Mueang District, Nonthaburi Province, Postal Code 11000 Tel. 02 9510000 Ext. 99655

https://www3.dmsc.moph.go.th/page-view/71

Point 6:  Figure 1: include a title, Clarify numbers so they can be easily tracked back to the figures provided at the stem of the design (ie, 120 patients and 234 controls). I'd suggest to merge Fig 1 and 2 in a unique flowchart.

Response 6: We edited by merge Fig 1 and 2 in a unique flowchart

The title of figure 1 is inserted into the figure 1 legend. “Enrollment diagram of the study population”.

Point 7: Figure 2 also lacks a title. 

Response 7: The title of figure 2 is inserted into the legend. “Violin plots comparing level of IgG anti-SRBD titer in CoronaVac-ChAdOx1 group at 2 and 4 weeks”.

Point 8: Results: From April to August 2021 or rather May to August as stated in the abstract?

Response 8: We edited the text for consistency, the study conducted between May to August 2021.  line: 52 in Abstract section line: 281 in Result section The study conducted between May to August 2021

Ponit 9: Figure 7 lacks a title. 

Response 9: The title of figure 7 is inserted into the legend. “Correlation between PVNT50 titer with Ab and anti-SRBD level by virus variant group”.

Ponit 10: Discussion: please, mention whether the sample size do limit,

beside AE conclusions, conclusions overall. Given that the Thailand national strategy mostly consisted on the IV-AV regimen from a certain point in time onwards, the cohort could have been easily quite larger.

Response 10: Sample size

Based on preliminary similar studies in Thailand at the mid-2021(18), we estimated that power of the sample size in this study

using the following parameters: 3 groups (including IV-AV, IV-IV, AV-AV), effect size = 0.3299, significant level = 95% and power = 80% using chi square test on R software. We would like to assure that the minimum sample size was at least 89 participants (N =30 each group with 10% drop-out), at 80% power.

in the line 430-441 in discuss section

The findings of two studied COVID-19 vaccine regimens in Thailand19,20 showed that the probability of local and systemic AEs in IV-AV-vaccinated individuals seemed to be higher than in other groups. Although some individuals developed mild to moderate symptoms within a few days after vaccination, there were no serious adverse events found. The findings in this study are consistent with two earlier studies in Thailand19,20. Moreover, based on results from five publications, including this study, the World Health Organization (WHO) launched an interim recommendation for heterologous COVID-19 vaccination30. This suggests that policymakers should consider the implementation of heterologous COVID-19 vaccination in the country depending on the safety, vaccine efficacy, and vaccine supply

Point 11: Discussion: would you care commenting on the duration of immunogenicity effects in the different groups?

Response 11: We edited the line 481-500 to include this comment.

This study also has limitations in terms of the data duration of immunogenicity effects in different groups and efficacy against non-Delta variants. Regarding the efficacy of immunogenicity against other VOCs such as Delta and Omicron, several studies demonstrated that homologous and heterologous vaccination regimens seem to have lower efficacy against the Delta and Omicron variants33,34. Recently, some studies on the CoronaVac- ChAdOx1 regimen in Thailand have indicated that the duration of protection is between 16 and 20 weeks after receiving the second dose. Therefore, in Thailand, we suggest that individuals who received heterologous vaccination should get the booster dose 16-20 weeks after the second dose19. Based on a recent study in Finland, we estimated the duration of immunogenicity of the homologous ChAdOx1 regimen to be about 6 months35. Studies in healthcare workers who received homologous ChAdOx1 vaccination showed a VE of 89% at three months and 63% at six months following the second dose of the COVID-19 vaccine.35. According to other studies, breakthrough infection on the homologous. CoronaVac regimen was detected at approximately 88 days (IQR 68–100) after the second dosage36. As a result, Thailand recommends that people who receive the homologous regimens receive a booster dose after the second dosage at three months and six months, respectively.

Point 12: Discussion: could you speculate and mention recent studies on the efficacy of the assessed vaccines and the combined regimen on non-Delta variants?

Response 12: This study also has limitations in terms of the efficacy against non-Delta variants. Regarding the efficacy of immunogenicity against other VOCs such as Delta and Omicron, several studies demonstrated that two doses homologous and heterologous vaccination regimens seem to have lower efficacy against the Delta and Omicron variants33,34.

Check meaning retained.

Reviewer 2 Report

Mahasirimongkol and colleagues seek to provide a snapshot of clinical data to highlight the need for using a prime-boost vaccine regimen with the inactivated whole virus vaccine CoronoaVac (IV) and the adenoviral vector vaccine ChAdOx1 nCoV-19, AstraZeneca (AV) in response to the SARS-CoV-2 Delta Variant. This central question is of clear significance and is of immediate relevance to the on-going COVID-19 pandemic. The authors provide data indicating that the IV IV-AV regimen could induce higher immunogenicity and safety than the other evaluated regimens (IV-IV, AV-AV), and that it might be use against the Delta variant. However, it is unclear if this defense is adequately supported by the number of individuals recruited in each group (IV-AV group n= 155; IV-IV group N= 32; AV-AV group n= 47, CS group n= 120). 

English need extensive revision. Many paragraphs are difficult to understand. Readibility of the mansucript should be extensively improved.

Author Response

Response to Reviewer 2 Comments

Point 1: Mahasirimongkol and colleagues seek to provide a snapshot of clinical data to highlight the need for using a prime-boost vaccine regimen with the inactivated whole virus vaccine CoronoaVac (IV) and the adenoviral vector vaccine ChAdOx1 nCoV-19, AstraZeneca (AV) in response to the SARS-CoV-2 Delta Variant. This central question is of clear significance and is of immediate relevance to the on-going COVID-19 pandemic. The authors provide data indicating that the  IV-AV regimen could induce higher immunogenicity and safety than the other evaluated regimens (IV-IV, AV-AV), and that it might be use against the Delta variant. However, it is unclear if this defense is adequately supported by the number of individuals recruited in each group (IV-AV group n= 155; IV-IV group N= 32; AV-AV group n= 47, CS group n= 120). 

Response 1: As mentioned about sample size in Reviewer2's Comment, we roughly estimated the total sample size which should be at least 56 participants or 14 per each group using library (pwr) on R software. We calculated the sample size based on effect size from study (19)  which likely indicated that the heterologous regimens in the Thai population had a larger effect size than other homologous regimens. Therefore, We estimated effect size = 0.5, at 0.05 significance level and 80% power. The sample sizes in this study is the largest for IV-AV group, to support the recommendation of the IV-AV group in Thailand.

Reviewer 3 Report

Study was performed in the small group and do not include or even discuss possible protection against Omicron variant, which is currently the crucial issue.

Author Response

Response to Reviewer 3 Comments

Point 1: Study was performed in the small group and do not include or even discuss possible protection against Omicron variant, which is currently the crucial issue.

Response 1: add in the line 481-500 in limitation section

This study also has limitations in terms of the data duration of immunogenicity effects in different groups and efficacy against non-Delta variants. Regarding the efficacy of immunogenicity against other VOCs such as Delta and Omicron, several studies demonstrated that homologous and heterologous vaccination regimens seem to have lower efficacy against the Delta and Omicron variants33,34. Recently, some studies on the IV-AV regimen in Thailand have indicated that the duration of protection is between 16 and 20 weeks after receiving the second dose. Therefore, in Thailand, we suggest that individuals who received heterologous vaccination should get the booster dose 16-20 weeks after the second dose19. Based on a recent study in Finland, we estimated the duration of immunogenicity of the AV-AV regimen to be about 6 months35 . Studies in healthcare workers who received AV-AV vaccination showed a VE of 89% at three months and 63% at six months following the second dose of the COVID-19 vaccine35. According to other studies, breakthrough infection on the IV-IV regimen was detected at approximately 88 days (IQR 68–100) after the second dosage36. As a result, Thailand recommends that people who receive the homologous regimens receive a booster dose after the second dosage at three months and six months, respectively

Round 2

Reviewer 1 Report

The authors have satisfactorily addressed my questions and remarks
